# Basic Tool Design Guidelines for Friction Stir Welding of Aluminum Alloys

**Elizabeth Hoyos *** and **María Camila Serna**

Department of Mechanical Engineering, Universidad EIA, Envigado 055428, Colombia; maria.serna14@eia.edu.co
* Correspondence: elizabeth.hoyos@eia.edu.co

**Abstract:** Friction Stir Welding (FSW) is a solid-state welding process that has multiple advantages over fusion welding. The design of tools for the FSW process is a factor of interest, considering its fundamental role in obtaining sound welds. There are some commercially available alternatives for FSW tools, but unlike conventional fusion welding consumables, their use is limited to very specific conditions. In this work, equations to act as guidelines in the design process for FSW tools are proposed for the 2XXX, 5XXX, 6XXX, and 7XXX aluminum series and any given thickness to determine: pin length, pin diameter, and shoulder diameter. Over 80 sources and 200 tests were used and detailed to generate these expressions. As a verification approach, successful welds by authors outside the scope of the original review and the tools used were evaluated under this development and used as case studies or verification for the guidelines. Variations between designs made using the guidelines and those reported by other researchers remain under 21%.

**Keywords:** FSW; aluminum alloys; pin; shoulder

## 1. Introduction

Friction Stir Welding (FSW) is a solid-state welding process patented in 1991 by The Welding Institute (TWI). This method is performed by utilizing a non-consumable cylindrical tool that rotates and advances in the material to be welded; this movement produces heat through friction and mixes the softened material to produce the weld [1]. Aluminum alloys are the second most used metal after steel, due to their high strength-to-weight ratio and thermal and electrical conductivities [2]. Annually, FSW applications increase due to the excellent results obtained with these alloys [3]. They are used in the railway [4], aerospace [5,6], automotive [7,8], and shipbuilding industries [9]. Table 1 shows the different types of aluminum alloys that are commercially available, along with basic conventional applications.

For FSW, the use of a tool is required, which plays an important role in the process [10] and consists of a shoulder and a pin, both playing a crucial role in the welding process. The shoulder is responsible for generating much of the heat required and the pin is responsible for transporting the plasticized material [1]. The tool contributes to the joint soundness since it directly impacts factors such as grain size, microstructure uniformity, and the way the material flows through the joint. The importance of the tool can be observed in Table 2, where the sum of the associated factors of it correspond to 75% [11].

Over the years, different tool features have been developed. Figures 1 and 2 show some of the various pin and shoulder designs reported. Along with choosing particular geometries, it is also necessary to select the pin length and both diameters. Selection is based on the specific application, thicknesses and materials to be welded, to name a few variables. Due to the reasons above, the tools are typically tailor made. The criteria used to define these characteristics are based on trial and error and it can be challenging and costly to develop cost-effective tools [12].

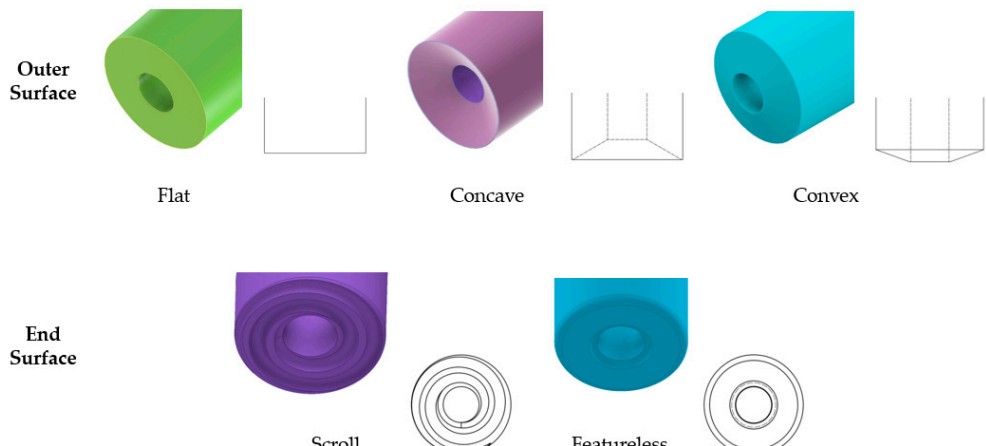

**Figure 1.** Types of shoulders.

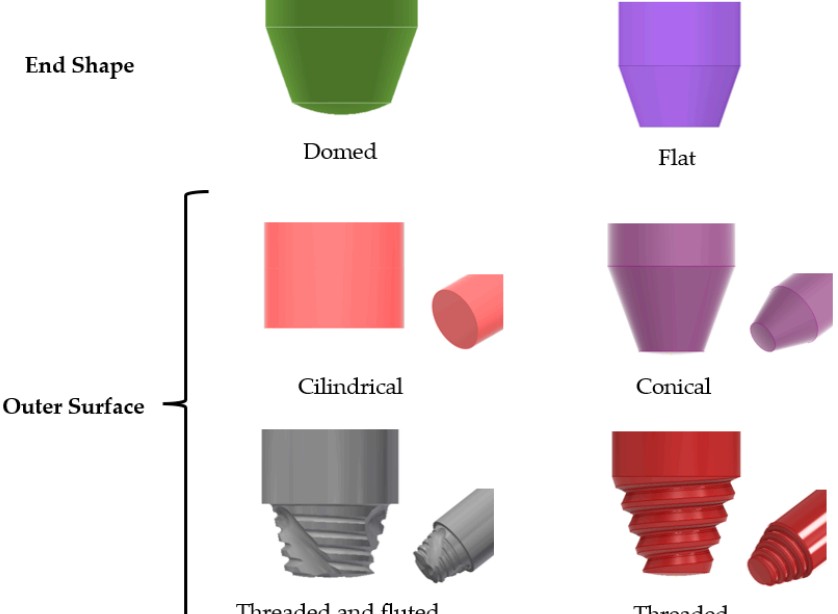

**Figure 2.** Types of pins.

**Table 1.** Aluminum alloys.

| Aluminum Alloys | Alloying Element | Applications |
| --- | --- | --- |
| 2XXX | Alloys in which copper is the principal alloying element [13], other alloys can be specified such as magnesium, silicon, manganese and iron [2] | Structural applications due to their good mechanical properties [14] |
| 5XXX | Alloys in which magnesium is the principal alloying element [13] | Automotive and electronic applications [15,16] |
| 6XXX | Magnesium and silicon are the principal aluminum alloys [2] | |
| 7XXX | Alloys in which zinc is the principal alloying element [13] | Automotive and electronic applications [15,17], aerospace industry (aircraft frame, spars and stringers) [18] |

**Table 2.** Percentage of butt joints parameters.

| Parameter | Percentage |
|---|---|
| Rotational speed | 5% |
| Travel speed | 5% |
| Tilt angle | 4% |
| Pin penetration | 28% |
| Shoulder/pin diameter ratio | 14% |
| Rotational speed and pin penetration interaction | 9% |
| Rotational speed and shoulder/pin diameter ratio interaction | 8% |
| Travel speed and shoulder/pin diameter ratio interaction | 15% |

Seeking to identify patterns in tool design, authors such as Y. N. Zhang [19] have made a compilation of the different characteristics such as shoulder and pin geometries. El-Moayed et al. [20] took a sample of 30 different published articles and made a review of the geometry of the tools used to then propose equations to determine the shoulder and pin diameter.

Sevvel et al. [21] made welds with different shoulders to classify the welds; in this way, they determined that the best D/T (shoulder diameter/thickness) ratio is 3.5. Authors such as M. Mehta et al. [22] showed, by trial and error, that the most important geometrical parameter in FSW tool design is the shoulder diameter. Tozaki et al. [23] tested different pin lengths to examine the effect of that parameter on weld microstructure.

The cited research shows the importance of the development of FSW tools. This article aims to guide the selection of some of the basic dimensions of conventional tools, for different aluminum alloys and plate thicknesses based on the data collected. It should be noted that in the literature there are not many approaches to tool design; most of the cases of tool selection involve intuition and experience [22]. It should be mentioned that FSW has multiple variants, such as bobbin and hybrid [24], which make use of tools with different configurations [25]. The equations proposed do not cover these cases.

## 2. Definition of Guidelines by Design Parameter

Through a bibliographic compilation, including 87 authors reporting 216 welds made by FSW in aluminum series 2XXX, 5XXX, 6XXX and 7XXX, with a butt joint configuration [26–110], the following list presents the different variables considered:

- Aluminum series;
- Rotational speed (rpm);
- Travel speed (mm/min);
- Angle (°);
- Pin diameter (mm);
- Shoulder diameter (mm);
- Pin type;
- Pin length;
- Shoulder type;
- Weld efficiency;
- Publication year.

Using this list, graphs were made to generate guidelines and patterns that facilitate the designing process for FSW tools in order to minimize the amount of trial and error [18]. Due to the amount of data, it was decided to average each variable according to the thickness of the base material. For example, for a material thickness of 2 mm, the trial results were pin diameters of 11.5, 10.5, 10, and 12 mm, and the final value of the pin diameter considered was 11 mm.

The results reported below were analyzed using a coefficient of determination ($R^2$), which indicates the relationship that the variables had; if $R^2$ is equal to 1 it corresponds to a perfect fit [111]. It should be noted that, for this study, an $R^2$ greater than 0.9 is considered acceptable and, as can be seen, all the graphs in Figures 3–8 have admissible values.

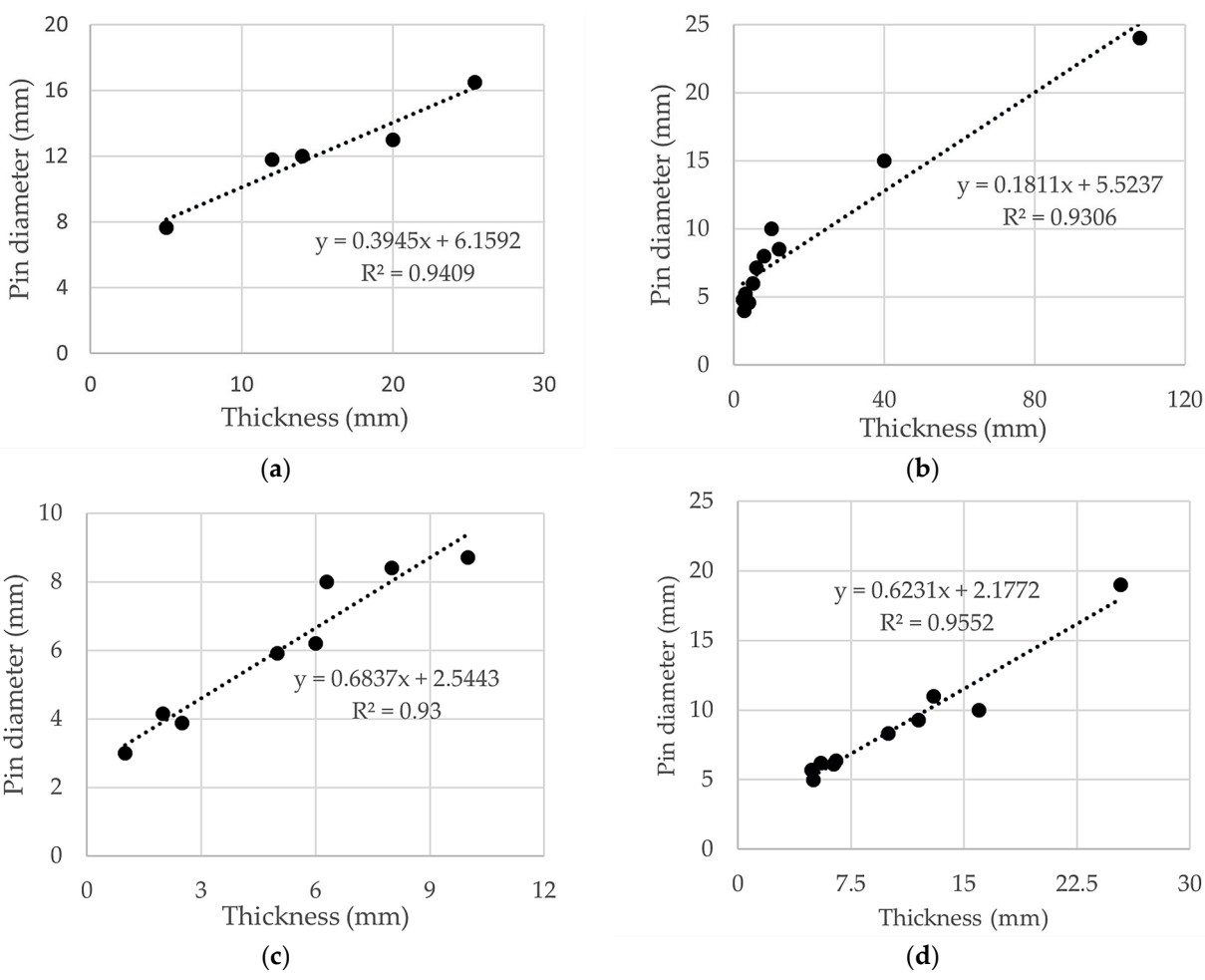

**Figure 3.** Pin diameter vs. thickness for series: (**a**) 2XXX; (**b**) 5XXX; (**c**) 6XXX, and (**d**) 7XXX.

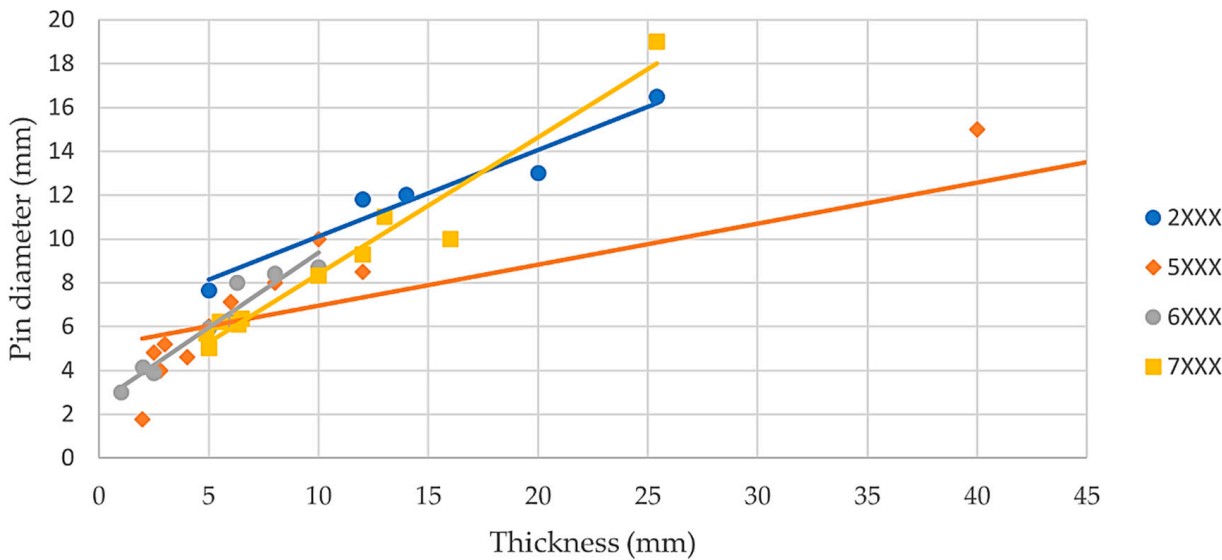

**Figure 4.** Summary of trend lines for pin diameter vs. thickness.

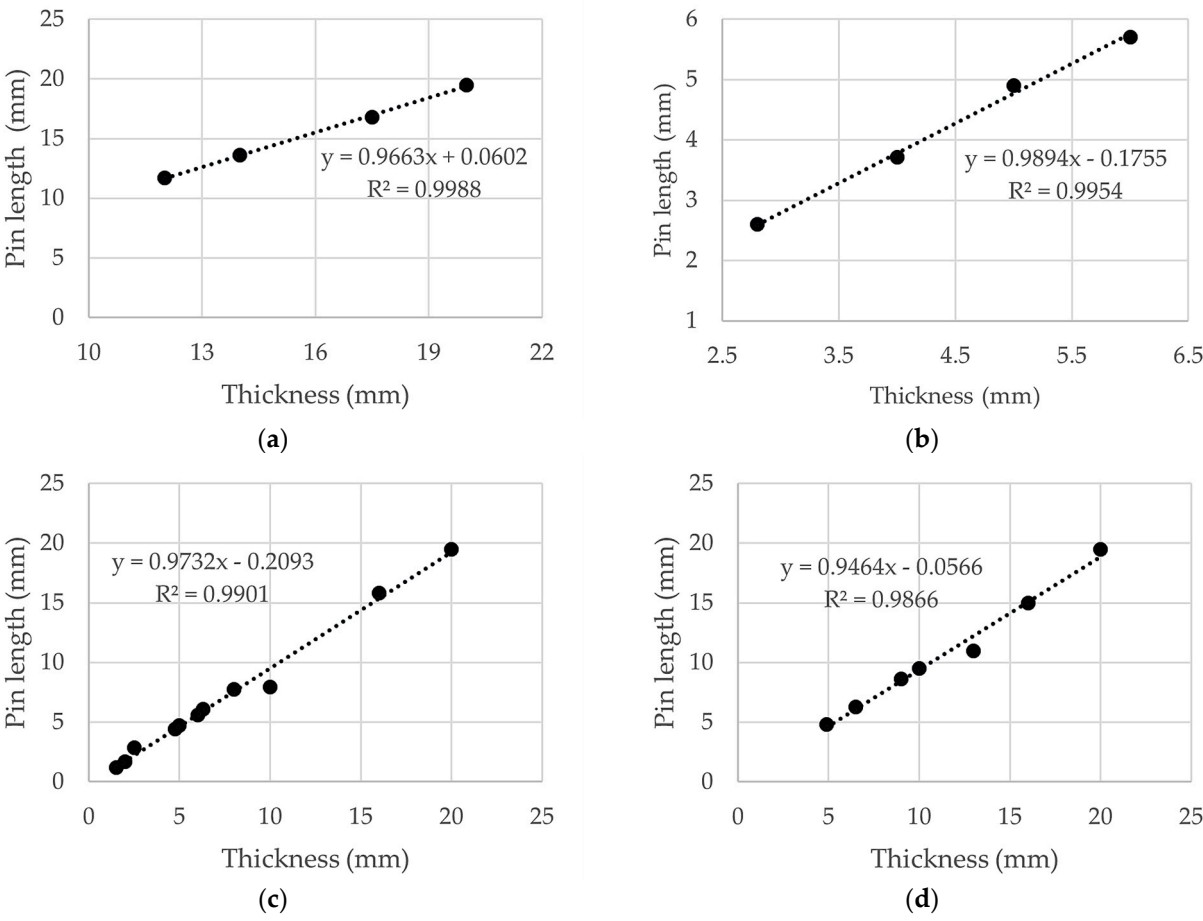

**Figure 5.** Pin length vs. thickness for series: (**a**) 2XXX; (**b**) 5XXX; (**c**) 6XXX; and (**d**) 7XXX.

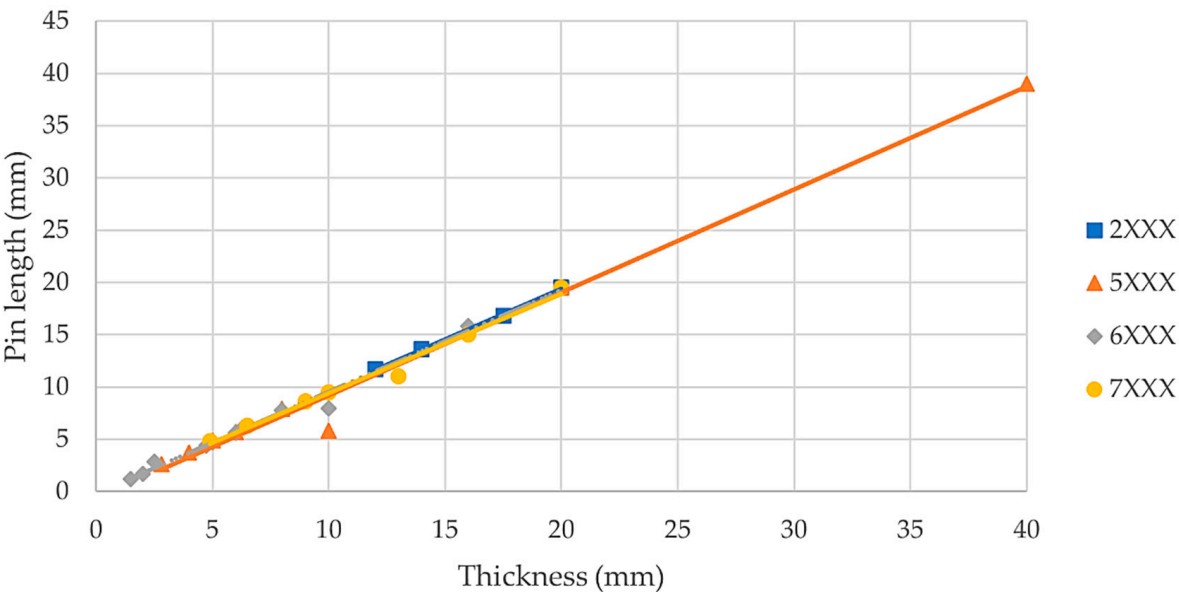

**Figure 6.** Summary of trend lines for pin length vs. thickness.

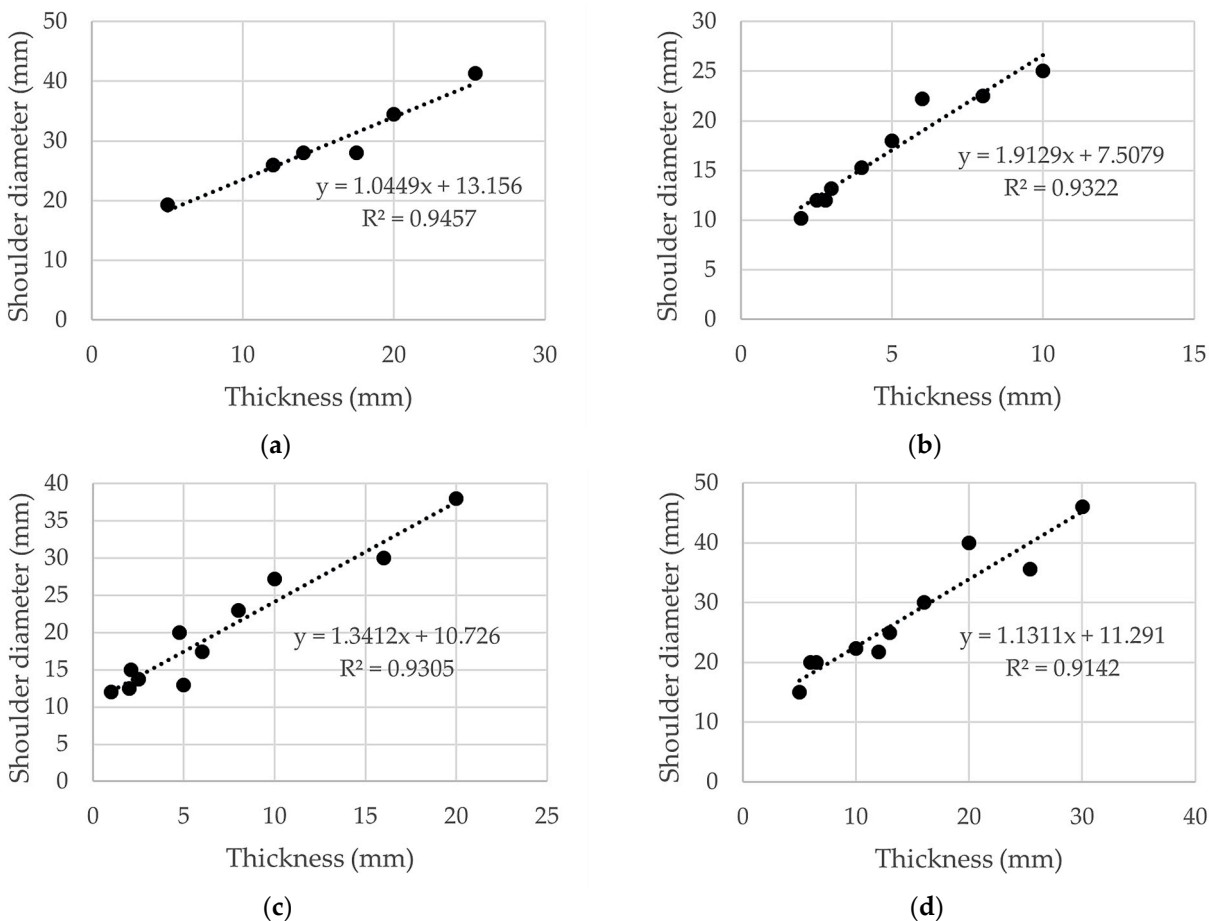

**Figure 7.** Shoulder diameter vs. thickness for series: (**a**) 2XXX; (**b**) 5XXX; (**c**) 6XXX; and (**d**) 7XXX.

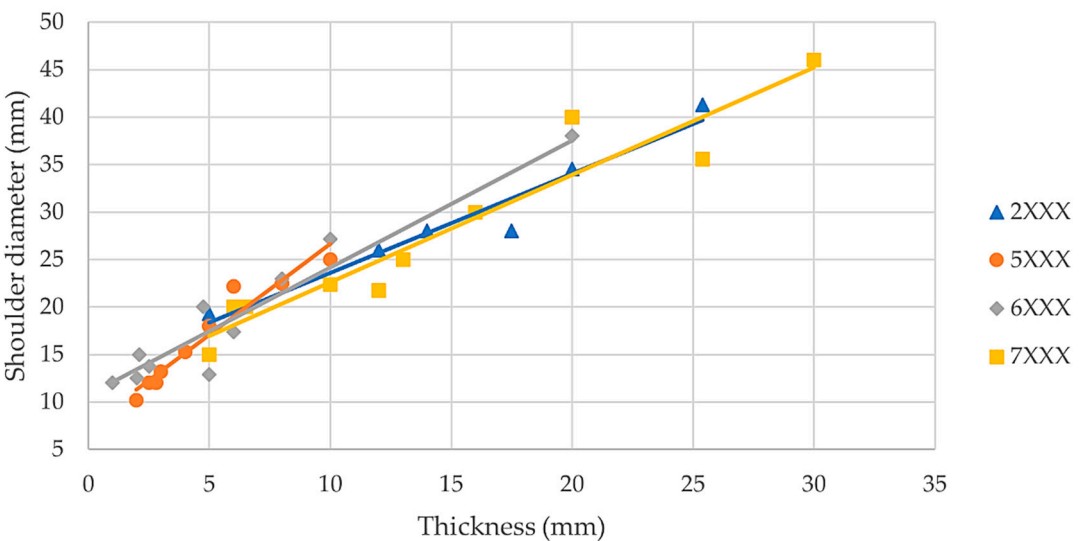

**Figure 8.** Summary of trend lines for shoulder diameter vs. thickness.

### 2.1. Pin Diameter

The pin is in charge of transporting the plasticized material along the joint [112]. Figure 3a–d present the pin diameter relative to material thickness; if the trial reported used a conical pin, the trend line included the largest diameter. It should be noted that the lowest $R^2$ was 0.93, so these graphs are considered to be within the established limits. Figure 7 shows a summary of the trend lines.

### 2.2. Pin Length

According to Wronska et al. [113] the pin has a key effect on the microstructural changes in the weld and thus impacts the strength of FSW joints. Pin length is usually estimated in tool design based on achieving full penetration, making plate thickness its essential variable [23]. Figure 5a–d have coefficients of determination ($R^2$) of 0.99 or higher, which means that the results comply with the defined criteria.

Figure 6 shows a summary of the trend lines presented previously; the overlapped slopes indicate that pin length does not depend on the type of alloy, but mainly on the thickness of the material to be welded. According to these, the difference between the pin length and material thickness should be kept between 5 and 6%, regardless of the aluminum series.

### 2.3. Shoulder Diameter

The shoulder is in contact with the surface during welding, and its function is to keep the material in position and generate most of the heat produced during welding [21]. Figure 7a–d show, according to the series, the trend line for the shoulder diameter resulting from the literature collection performed. Figure 8 is a summary of the shoulder diameter trend lines for each series. According to the trendlines, the minimum coefficient of determination was 0.9142 and the maximum was 0.9457, which are acceptable according to the threshold set.

In summary, the equations mentioned previously are shown in Table 3.

**Table 3.** Summary of equations by aluminum series and tool parameter.

| Series | Tool Feature | Equation |
|--------|-------------|----------|
| 2XXX | Shoulder diameter | y = 1.0449x + 13.156 |
| | Pin diameter | y = 0.3945x + 6.1592 |
| | Pin length | y = 0.9663x + 0.0602 |
| 5XXX | Shoulder diameter | y = 1.9129x + 7.5079 |
| | Pin diameter | y = 0.1811x + 5.5237 |
| | Pin length | y = 0.9894x − 0.1755 |
| 6XXX | Shoulder diameter | y = 1.3412x + 10.726 |
| | Pin diameter | y = 0.6837x + 2.5443 |
| | Pin length | y = 0.9732x − 0.2093 |
| 7XXX | Shoulder diameter | y = 1.1311x + 11.291 |
| | Pin diameter | y = 0.6231x + 2.1772 |
| | Pin length | y = 0.9464x − 0.0566 |

## 3. Results

### 3.1. Tool Design Example

To test the expressions previously developed, a tool was designed to weld a 6XXX series aluminum, specifically, AA 6061-T6, with a 6.5 mm thickness. The tool dimensions are proposed in Table 4. It is important to clarify that the expressions proposed only account for the basic tool dimensions; other aspects such as threading, pin shape, shoulder features, among others, were defined using trends identified in the literature review. For these characteristics, no expressions were proposed in this work as they did not exceed the threshold established for the coefficient of determination.

**Table 4.** Proposed dimensions for a tool using the suggested guidelines.

| Tool Parameter | Dimension (mm) |
|----------------|----------------|
| Shoulder diameter | 19 |
| Pin diameter | 6.5 |
| Pin length | 4.5 |

Using the information in Table 4, an AISI H13 tool with a removable pin was made. Figure 9a shows the proposed pin, and Figure 9b the shoulder design. A scroll, whose main advantage is that no tilt angle is required, was included as well.

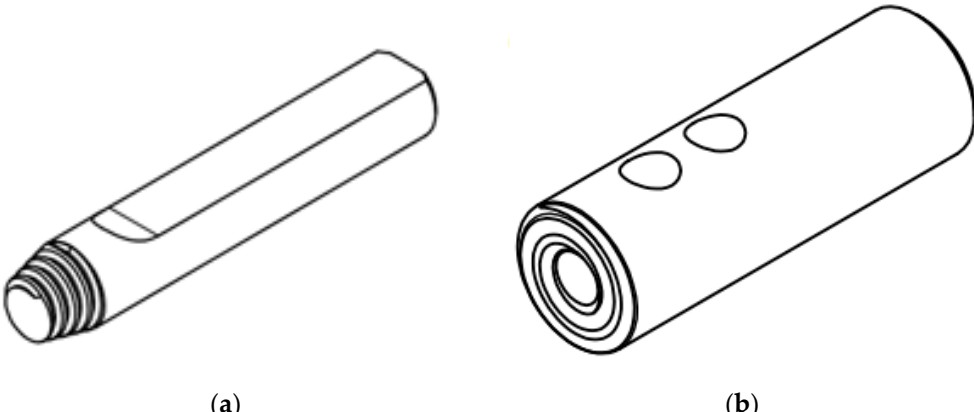

|          (a)          |          (b)          |

**Figure 9.** Tool design: (**a**) pin; (**b**) shoulder.

### 3.2. Validation through Experiments by Authors Outside the Initial Review

For verification purposes, comparisons were made with various tests carried out by authors outside the initial review [114–117]. The basic dimensions of FSW tools were calculated, according to the material and thickness to be welded, and compared with those used by the researchers and deemed as adequate, in some cases using the efficiency of the joints. Efficiency is defined as the strength of a welded joint with respect to the strength of the base metal [118]. Due to its material dependence, each researcher proposed their acceptable efficiency [119]. Tables 5–8 show the comparison between the of results tool design guidelines and experimental work. The "objective" column values were obtained using the expression in the column called "equation"; thus, in each case, the x was replaced by the thickness used in the test, and the column designated with the name "real" corresponds to the dimensions used in each experimental work.

### 3.2.1. Serie 2XXX

The study carried out by Z. Zhang, B. L. Xiao and Z. Y. Ma, used Al 2219-T6 plates, which were 5.6 mm thick and reached an efficiency of 79%; in their work, an acceptable efficiency starts at 65% [114]. Table 5 shows the variations for shoulder diameter, pin diameter and pin length, and the highest error obtained was 5%.

**Table 5.** Comparison between tool design guidelines and experimental work by other authors for series 2XXX.

| Tool Feature | Equation | Objective | Real | Error |
|:---:|:---:|:---:|:---:|:---:|
| Shoulder diameter | y = 1.0449x + 13.156 | 19.01 | 20 | 4.96% |
| Pin diameter | y = 0.3945x + 6.1592 | 8.37 | 8 | 4.61% |
| Pin length | y = 0.9663x + 0.0602 | 5.47 | 5.4 | 1.32% |

**Table 6.** Comparison between tool design guidelines and experimental work by other authors for series 5XXX.

| Tool Feature | Equation | Objective | Real | Error |
|:---:|:---:|:---:|:---:|:---:|
| Shoulder diameter | y = 1.9129x + 7.5079 | 17.07 | 15 | 13.82% |
| Pin diameter | y = 0.1811x + 5.5237 | 6.43 | 6 | 7.15% |
| Pin length | y = 0.9894x − 0.1755 | 4.77 | 4.5 | 6.03% |

**Table 7.** Comparison between tool design guidelines and experimental work by other authors for series 6XXX.

| Tool Feature | Equation | Objective | Real | Error |
|:---:|:---:|:---:|:---:|:---:|
| Shoulder diameter | y = 1.3412x + 10.726 | 19.43 | 16 | 21.44% |
| Pin diameter | y = 0.6837x + 2.5443 | 6.99 | 8 | 12.65% |
| Pin length | y = 0.9732x − 0.2093 | 6.12 | 5.8 | 5.46% |

**Table 8.** Comparison between tool design guidelines and experimental work by other authors for series 7XXX.

| Tool Feature | Equation | Objective | Real | Error |
|:---:|:---:|:---:|:---:|:---:|
| Shoulder diameter | y = 1.1311x + 11.291 | 16.7 | 20 | 16.6% |
| Pin diameter | y = 0.6231x + 2.1772 | 5.1 | 6 | 14.3% |
| Pin length | y = 0.9464x − 0.0566 | 4.5 | 5 | 11.0% |

### 3.2.2. Serie 5XXX

The paper *"The effects of processing environments on the microstructure and mechanical properties of the Ti/5083Al composites produced by friction stir processing"* shows different trials with 5 mm thick Al 5083 [115]. Table 6 shows the error resulting in the comparison between calculated and experimentally verified tools; the maximum was 13.82%.

### 3.2.3. Serie 6XXX

The purpose of the previous work titled *"Implementation of Friction Stir Welding (FSW) in the Colombian rail transport sector"* was to weld a piece of "Metro de Medellín" that had 6.5 mm thickness and AA 6082-T6 material [116]. It should be noticed that this comparison has the biggest error (21.44%), which is for the shoulder diameter. It is worth mentioning that the shoulder diameter selected in this application obeys a specific aspect of the geometry of the part to be welded, which limited its dimensions.

### 3.2.4. Serie 7XXX

The tests carried out were made with 7075-T6 aluminum and a plate thickness of 3/16 in (4.8 mm approximately) [117], with a minimum efficiency of 60%, observing the AWS D17.3 code for a 6XXX series with T6 tempering [120]. An X-ray of the weld can be seen in Figure 10, and Table 8 shows the comparison between the experiment and the basic tool design equations proposed.

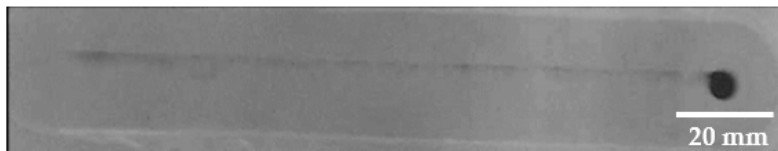

**Figure 10.** X-ray of an AA7075-T6 aluminum FSW weld [117].

## 4. Welding Experimental Validation

To validate the equations proposed previously for different aluminum alloys (Table 3), welds were performed with 3/16″ thick AA 6061-T6 aluminum. The test plate dimensions are presented in Figure 11. Table 9 shows the mechanical properties of the material. Different tools were used for each of the welds, and only different pin lengths and other dimensions (shoulder and pin diameter) were preserved. According to this, Tool 1 had an 18.1 mm shoulder diameter; the pin diameter was 6.4 mm and the pin length was 4.4 mm (view Figure 12). All dimensions were calculated with the equations of Table 3. Tool 2 had

a pin length of 2.2 mm and underwent a 50% reduction. A rotational speed of 650 rpm and a travel speed of 45 mm/min were the parameters used for the welds.

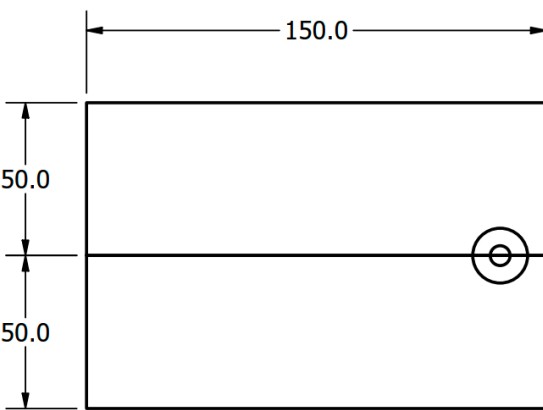

**Figure 11.** Test plate dimensions (all units in mm).

**Table 9.** Mechanical properties of Al 6061-T6 [121].

| Base Material | Microhardness, HV | UTS, Mpa | Yield Strength (Mpa) | Elongation (%) |
|---|---|---|---|---|
| Al 6061-T6 | 107 | 290 | 255 | 12 |

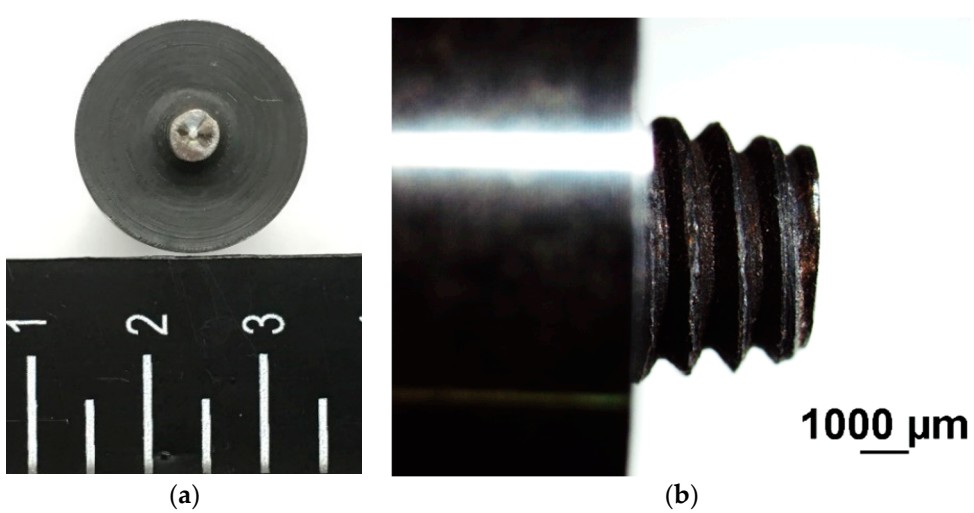

**Figure 12.** Tool 1 design: (**a**) shoulder and (**b**) pin.

*4.1. Non-Destructive Tests (NDT)*

Non-destructive tests were employed to verify the test weld soundness as follows.

4.1.1. X-rays

Radiography tests are non-destructive and use electromagnetic radiation with wavelengths shorter than those of ultraviolet light [122]. Figure 13 shows the X-ray corresponding to the weld made with Tool 1, and it can be said that the weld has no volumetric discontinuities; therefore, it is a sound weld.

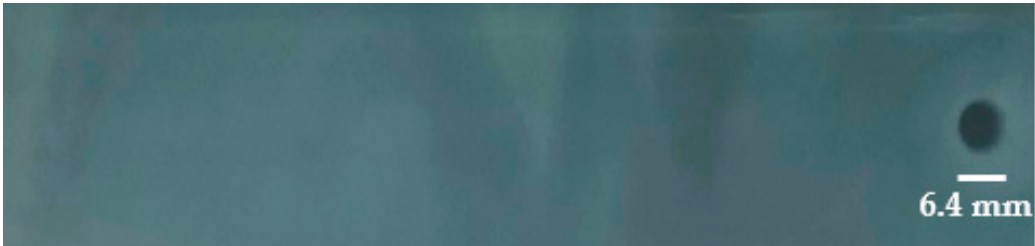

**Figure 13.** Tool 1 trial—X-ray of an AA 6061-T6 aluminum FSW weld.

The X-ray results according to trial 2 are shown in Figure 14, indicating a discontinuity (box in red).

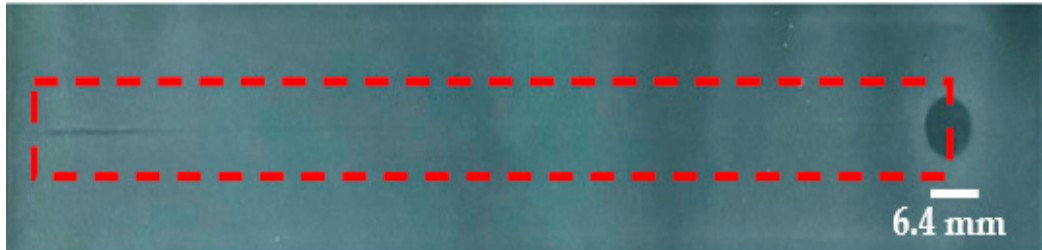

**Figure 14.** Tool 2 trial—X-ray of an AA 6061-T6 aluminum FSW weld.

### 4.1.2. Ultrasound

According to NDT Resource Center, ultrasound tests are non-destructive and use ultrasonic waves to create an image of the inside of an object [123]. The ultrasonic test performed for Tool 2 results (view Figure 15) indicated a cavity along the weld (view Figure 16). The ultrasound obtained for a weld made using Tool 1 does not show any indication of volumetric discontinuities.

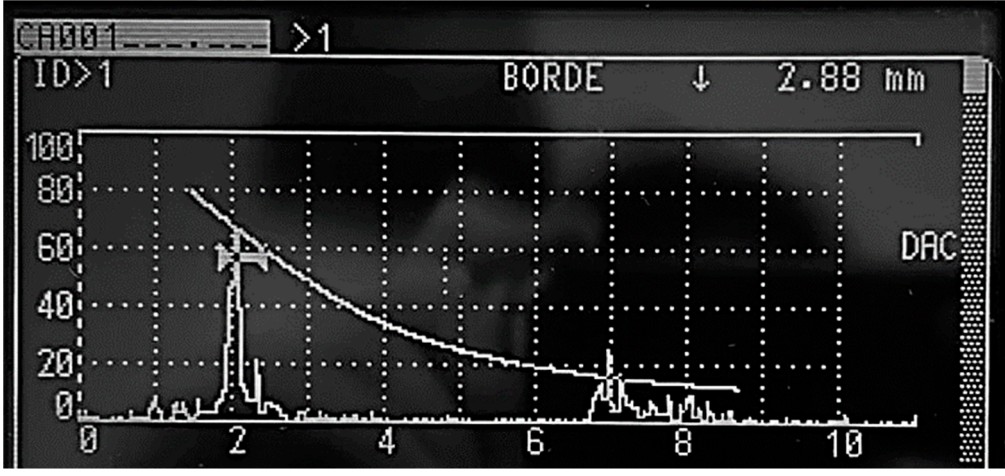

**Figure 15.** Ultrasound results with indication for Tool 2 trial (EPOCH 4 ultrasound system).

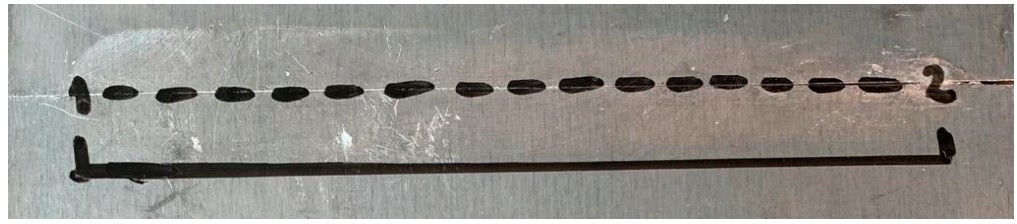

**Figure 16.** Cavity location for Tool 2 trial, according to ultrasound results (EPOCH 4 ultrasound system).

By considering the results of the non-destructive tests, it can be concluded that the equations developed and used for dimensioning FSW Tool 1 can be useful. No discontinuities were found in the weld made with Tool 1; on the other hand, Tool 2, which has non-corresponding dimensions and was used with the same welding parameters, presented major discontinuities that can be observed in Figures 15 and 16. It is clear that there are multiple causes of potential discontinuities and failure in general for FSW, so the proposed exercise can be expanded using direct experimentation and a bibliographic review that considers additional tool variations than those used in this work.

## 5. Conclusions

FSW tool design requires the consideration of various factors and involves multiple features to be defined. The results from this work allow obtaining basic tool dimensions that serve as a first step in design, based on the thickness to be welded and the series of aluminum used. Other factors such as pin shape, shoulder design, and whether or not an inter-changeable pin is used are at the discretion of the designer. As mentioned previously, aspects such as these can be defined using the trends identified in the literature review. However, in this work, no expressions were proposed since the coefficient of determination found in their analyses did not exceed the threshold established. The collection of more data in the future could allow this additional progress.

Some interesting aspects to consider are that the length of the pin does not depend on the aluminum series but mainly on the thickness of the material to be welded; also, the difference between the length of the pin and the thickness should be kept between 5 and 6%. The 5XXX series requires smaller shoulder and pin diameters than the 2XXX, 6XXX and 7XXX series. Similar shoulder diameters are used for series 2XXX and 7XXX.

The verifications carried out using the successful tools reported by researchers, outside the sources initially used, considered for the design of the guidelines, have variations in dimensions between 0 and 21.44%, although this high value can be explained considering the specific space restrictions of the part being welded. Additionally, the tests carried out with the tool manufactured using the proposed guidelines generated sound welds after being evaluated using X-rays and ultrasound.

**Author Contributions:** Conceptualization, E.H. and M.C.S.; methodology, E.H. and M.C.S.; validation, E.H. and M.C.S.; formal analysis, E.H. and M.C.S.; investigation, M.C.S.; resources, E.H. and M.C.S.; data curation, E.H. and M.C.S.; writing—original draft preparation, E.H. and M.C.S.; writing—review and editing, E.H. and M.C.S.; supervision, E.H.; project administration, E.H.; funding acquisition, E.H. All authors have read and agreed to the published version of the manuscript.

**Funding:** This research project is by the Transforming Systems through Partnership (TSP) programme (TSP1094). Run by the Royal Academy of Engineering and supported by the Newton Fund.

**Institutional Review Board Statement:** Not applicable.

**Data Availability Statement:** Data is contained within the article.

**Conflicts of Interest:** The authors declare no conflict of interest.

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
