# Peer review of "Basic Tool Design Guidelines for Friction Stir Welding of Aluminum Alloys"

_metals, doi:10.3390/met11122042_

Round 1
Reviewer 1 Report
Dear Authors,
Your paper contains Many Grammatical Errors and Technical Errors. In addition to this, the Reference Nos mentioned inside the Manuscript mismatch with the Nos mentioned in the Reference Section.
Detailed Comments to the Authors are attached as a Separate Word Document. Authors are strongly advised to make all the necessary changes mentioned in the PDF Document.

Reviewer 2 Report
Some comments were listed below for a further improvement of the paper.
- The title of this manuscript is “Tool Design Guidelines for FSW”. But the design of FSWtolls is not only for pin diameter, pin length and shoulder diameter. So I think the title of this manuscript is not appropriate.
- All the references should be cited in order.
- There are a lot of figures, tables, and chapter number errors in the manuscript.
- From figure 3(b),(c), figure 7(b),(c), it seems that the linear fitting isn’t suitable for pin diameter and shoulder diameter for 5XXX and 6XXX series aluminum alloys.
- The FSW process parameters have great influence on the quality of weldment. How to judge the tool design is appropriate?
- In section 4, non-destructive tests were conducted to verify weld soundness. Mechanical properties of weldment should be completemented.
Reviewer 3 Report
The article attempts to derive FSW tool design relationships for aluminium alloys by analysing the available literature. The attempt is good, but the investigation has some shortcomings, as listed below:
- The investigation is about aluminium Aloys. The title gives an impression of a generic tool design for any material. The title should be revised with the addition of Aluminium alloys.
- Of the three design parameters, pin diameter, shoulder diameter, and pin length, only the first two affect The pin length is always fixed according to the depth of penetration required or the plate thickness (if full penetration is required). This is why the fit between pin length and the plate thickness is very good as pin length is decided based on the thickness. Section 2.1 should be removed, or the above-stated fact should be explicitly mentioned in support of the difference in the fit of pin length equations compared to the other two.
- The equations for 5XXX and 6XXX are or correct. Only a few samples of very high thickness are used, and they are acting as outliers. They are pushing the fit towards themselves (Fig. 3(b) and(c). This anomaly may be removed by either removing the high thickness data from the line fitting exercise or using a higher-order (2nd or 3rd order) fit.
- Similar to point 3, lines in 7(a) and 7(b) needs to be corrected. This change will improve the R2 value and bring the maximum error down from 22%.
- The validation is based on assumed efficiency, e.g. 60%. This efficiency should be justified with reported efficiency of should be mentioned as a limitation of this study. The argument behind the assumption of efficiency should be strengthened by explaining material dependence; for example, the efficiency may reach almost 100 % in other materials. See ‘Induction heated tool-assisted friction-stir welding (i-FSW): A novel hybrid process for joining thermoplastics’.
- The articles neglect the pin design while developing the equations. Also, several important tool designs like hybrid design are not mentioned; for example, see ‘Role of hybrid tool pin profile on enhancing welding speed and mechanical properties of AA2219-T6 friction stir welds’. This should be explained as a limitation of the study by mentioning different tool designs mentioned above and are not listed in Fig. 2.
- There are several versions of the FSW process and tool design aspects available in the literature. See, ‘ Evolution and Current Practices in Friction Stir Welding Tool Design, Advanced Welding and Deforming. The introduction section should mention them and explain which kind of process version this investigation is using.
- The coefficient in the equation uses (,) for decimal. Instead, (.) should be used.
- The article needs a thorough check of the language and grammatical errors.
Round 2
Reviewer 1 Report
In the Reference Section, Reference Nos:21 and 116 are the same Papers. i.e., the same paper "Sevvel, P.; Jaiganesh, V. Effect of Tool Shoulder Diameter to Plate Thickness Ratio on Mechanical Properties and 571 Nugget Zone Characteristics During FSW of Dissimilar Mg Alloys. Transactions of the Indian Institute of Metals 572 2015, 68, 41–46, doi:10.1007/s12666-015-0602-0" has been mentioned twice in the paper. Remove the reference No:116, as Ref No:21 is cited inside the Manuscript with description. Replace it with a suitable one by selecting a paper from the below mentioned list:-
- Dhanesh Babu, S.D., Sevvel, P, Senthil Kumar, R, Vijayan, V & Subramani, J: “Development of Thermo Mechanical Model for Prediction of Temperature Diffusion in Different FSW Tool Pin Geometries During Joining of AZ80A Mg Alloys”, Journal of Inorganic and Organometallic Polymers and Materialsa, 31 (7), 2021, 3196–3212. https://doi.org/10.1007/s10904-021-01931-4
- Stephan Thangaiah I S, Sevvel P, Satheesh C and Mahadevan S, “Experimental Study on the Role of Tool Geometry in determining the Strength & Soundness of Wrought AZ80a Mg Alloy Joints During FSW Process”, FME Transactions, Vol. 46, No.4, 612 – 622, 2018. https://doi.org/10.5937/fmet1804612T
- Sevvel, P &Jaiganesh, V: “Impact of Tool Profile on Mechanical Properties of AZ31B Mg Alloy during FSW Using Optimized Parameters”, FME Transactions, Vol. 44(1), 2016, pp. 43-49.https://doi.org/10.5937/fmet1601043J
- Jaiganesh, V, Sevvel, P & Nagarajan P.K: “Impact of Tool Pin Geometry and Optimized Process Parameters on Mechanical Properties of Friction Stir Welded AZ80A Mg Alloy”, Materials Science Forum, Vol. 866, 2016, pp. 151-155.https://doi.org/10.4028/www.scientific.net/MSF.866.151
Reviewer 2 Report
The manuscript has been revised by authors accroding to reviewers' questions.Reviewer 3 Report
All the comments are satisfactorily addressed. The manuscript may be accepted.